# A Tight Interaction between the Native Seagrass *Cymodocea nodosa* and the Exotic *Halophila stipulacea* in the Aegean Sea Highlights Seagrass Holobiont Variations

**DOI:** 10.3390/plants12020350

**Published:** 2023-01-11

**Authors:** Chiara Conte, Eugenia T. Apostolaki, Salvatrice Vizzini, Luciana Migliore

**Affiliations:** 1PhD Program in Evolutionary Biology and Ecology, University of Rome Tor Vergata, 00133 Rome, Italy; 2Laboratory of Ecology and Ecotoxicology, Department of Biology, University of Rome Tor Vergata, 00133 Rome, Italy; 3Institute of Oceanography, Hellenic Centre for Marine Research, P.O. Box 2214, 71003 Heraklion, Crete, Greece; 4Department of Earth and Marine Sciences, University of Palermo, Via Archirafi 18, 90123 Palermo, Italy; 5CoNISMa, National Interuniversity Consortium for Marine Sciences, Piazzale Flaminio 9, 00196 Roma, Italy; 6eCampus University, Via Isimbardi 10, 22060 Novedrate (CO), Italy

**Keywords:** seagrass interaction, microbiota, biological invasion, seagrass descriptors, Indicator Species Index

## Abstract

Seagrasses harbour bacterial communities with which they constitute a functional unit called *holobiont* that responds as a whole to environmental changes. Epiphytic bacterial communities rapidly respond to both biotic and abiotic factors, potentially contributing to the host fitness. The Lessepsian migrant *Halophila stipulacea* has a high phenotypical plasticity and harbours a highly diverse epiphytic bacterial community, which could support its invasiveness in the Mediterranean Sea. The current study aimed to evaluate the *Halophila/Cymodocea* competition in the Aegean Sea by analysing each of the two seagrasses in a meadow zone where these intermingled, as well as in their monospecific zones, at two depths. Differences in holobionts were evaluated using seagrass descriptors (morphometric, biochemical, elemental, and isotopic composition) to assess host changes, and 16S rRNA gene to identify bacterial community structure and composition. An Indicator Species Index was used to identify bacteria significantly associated with each host. In mixed meadows, native *C. nodosa* was shown to be affected by the presence of exotic *H. stipulacea*, in terms of both plant descriptors and bacterial communities, while *H. stipulacea* responded only to environmental factors rather than *C. nodosa* proximity. This study provided evidence of the competitive advantage of *H. stipulacea* on *C. nodosa* in the Aegean Sea and suggests the possible use of associated bacterial communities as an ecological seagrass descriptor.

## 1. Introduction

All organisms, including seagrasses, are colonised by different microbial communities, such as bacteria, fungi, and viruses, constituting a functional unit called holobiont [1,2,3]. The host establishes ephemeral or long-lasting interactions with the associated microbial communities [4,5], which can respond quickly to environmental changes by modifying their structure and composition [6,7,8]. Seagrasses and their bacterial communities may establish symbiotic relationships in which seagrasses provide microbes with a chemically different microenvironment and labile or recalcitrant organic matter [9,10,11,12]. In turn, seagrasses may exploit the microbial genetic/metabolic versatility, one of the critical points of the plant/bacteria association [13,14,15].

Recent studies reported the crucial role of bacteria in host ecophysiology. They contribute to a variety of processes, such as cycling and uptake of nutrients [12], rhizosphere detoxification from phytotoxic sulfide [15,16,17,18,19], and plant hormone-like compounds production [10], recently reviewed by Ugarelli et al. [9], Tarquinio et al. [13], and Conte et al. [20]. 

The complex epiphytic bacterial communities associated with seagrass leaves or roots/rhizomes are taxonomically different. Two bacterial components can be found in both plant parts: (i) a *bacterial core*, generally composed of the most represented taxa, related to broad-scale functions [21]; and (ii) *host-specific* bacteria, usually the less represented taxa, related to finer scales of ecosystem functioning, such as the host ecophysiology [20,21,22]. Consequently, both the environmental conditions and the seagrass ecological status may determine the enrichment of specific bacterial taxa within the community, and these changes may enhance host acclimatisation to changing conditions [7,14,23,24,25]. Thus, the investigation of seagrass–bacteria dynamics can widen our knowledge of the seagrass ecophysiology, unveiling seagrass’ response to different pressures and, in the long term, it may also represent an early warning seagrass descriptor [20,26,27,28].

*Cymodocea nodosa* is a dioecious seagrass occurring throughout the Mediterranean Sea and in the eastern Atlantic Ocean from Portugal to Senegal [29,30]. It lives in open waters on sandy and muddy substrates, but it can also colonize coastal lagoons, tolerating natural salinity fluctuations [29,30,31,32]. Little is known about *C. nodosa*’s epiphytic bacterial communities [25,33,34]. These studies showed that *C. nodosa* harbours a rich microbial community even under different pH changes [33] and maintains its host-specific microbial community when in syntopy with the invasive seaweed *Caulerpa cylindracea* [34].

*Halophila stipulacea* is a dioecious, small sub-tropical and tropical seagrass native to the Red Sea, the Persian Gulf, and the Indian Ocean [35]. It thrives under a wide range of environmental conditions, due to a high plasticity [36] that allows it to modify its phenotypical traits according to different bathymetric and eutrophic conditions, or temperature ranges [37,38,39,40,41,42]. It also shows high tolerance, resistance, and resilience to temperature shift, sulfide, natural disturbance [42,43,44,45]. *H. stipulacea* entered the Mediterranean Sea as a Lessepsian migrant at the end of the XIX century [46,47,48] and slowly spread, forming monospecific or mixed meadows with native seagrasses [47,48,49,50,51,52,53]. In the last two decades, instead, *H. stipulacea* reached and invaded the Caribbean Sea, rapidly displacing the native seagrasses (e.g., *Syringodium filiforme*) [54,55,56,57,58]. Bacterial communities associated with *H. stipulacea* have been extensively analysed in native areas, revealing a high variability in structure and composition, even at a small spatial scale [26,28,59,60,61,62]. Few studies investigated the *H. stipulacea* bacterial communities in invaded areas [22,63,64], showing a host-specific community in both the Caribbean and Mediterranean Seas. Nowadays, *H. stipulacea* keeps expanding in the rapidly warming Mediterranean Sea [51,52,53,65,66,67,68] and two studies hypothesized the possible competition with *C. nodosa* along the Tunisian coast [49,50,64], where a rapid increase in *H. stipulacea* density with a concurrent decrease in *C. nodosa* density has been reported. *H. stipulacea*–*C. nodosa* interaction in the Aegean Sea has not been investigated yet. 

In this study, the holobiont perspective (plant descriptors + bacterial community identification) has been applied to evaluate the interaction of the Mediterranean native seagrass *Cymodocea nodosa* (Ucria) Ascherson and the exotic *Halophila stipulacea* (Forsskål) Ascherson at two different depths. The study gives a first appraisal of the seagrasses’ holobiont variations, due to the putative interspecific competition, by evaluating: (i) host and associated bacteria’s response to abiotic and biotic pressures and (ii) bacterial response to the putative competition dynamics. In addition, the bacterial communities collected from the surrounding seawater and sediment were analysed to evaluate the bacterial pool in the areas [69].

## 2. Results

### 2.1. Abiotic Setting

The inorganic nutrients in bottom seawater showed comparable concentration in the two sites (*t*-Student test; *p* = NS) except for NO_3_, which was higher in site #1 (*t*-Student test = 3.86; *p* < 0.05; Table 1). 

Inorganic nutrients in pore water were comparable between sites (ANOVA, *p* = 0.05) and total nitrogen (TN%) in sediment was similar in the two sites; conversely, sediment organic carbon content (TC%), δ^13^C (‰) and δ^15^N (‰) were generally higher in the shallow site (#1) than in deep one (#2; Table 2). In both sites, the granulometric composition accounted for coarse/medium sand, being 70% above the fraction between 1000 and 250 μm, and no significant differences were found between the sites (PERMANOVA *p* = NS).

### 2.2. Biotic Setting

In site #1 (shallow), both species showed a higher meadow density compared to the site #2 (deep). This difference was significant only for *C. nodosa* in both meadow zones (Tukey’s pairwise post-hoc test; Table 3). Furthermore, in site #1, *C. nodosa* density was significantly higher in the monospecific meadow zone compared to the mixed one, while *H. stipulacea*’s density remain unchanged between meadow zones (Tukey’s pairwise post-hoc test; Table 3). Interestingly, *C. nodosa*’s meadow density significantly decreased with the increase in *H. stipulacea* density (R = −0.89, *p* < 0.01). *C. nodosa* was shown to have a better ecological status in site #1 compared to site #2, and in monospecific compared mixed meadows, as shown by the biometric and biochemical parameters. In fact, the *C. nodosa* leaf area was significantly higher in the monospecific than in the mixed meadow zone, and the total length of rhizomes (TLR) per square meter was longer in the shallow site (#1) compared to the deep one (#2), as well as higher in the monospecific meadow than the mixed one (Tukey’s pairwise post-hoc test; Table 3). Differently, *H. stipulacea* showed more evident changes between sites than meadow zones. The leaf area was significantly higher in the deep site (#2) compared to the shallow one (#1), without significant differences between monospecific and mixed zones (Tukey’s pairwise post-hoc test; Table 3). The total length of rhizomes (TLR) per square meter was higher in site #1 than site #2 (Tukey’s pairwise post-hoc test; Table 3). 

In both species, pigments’ content did not differ significantly between sites or meadow zones, except for Chl *b* content in both species from mixed zones (two-way ANOVA; Appendix A). Consequently, a significant reduction in Chl *a*/*b* ratio was observed between the sites (two-way ANOVA; Appendix A).

Total phenols followed the same pattern in both seagrass species. They increased in both species with the depth and the proximity to the other species (two-way ANOVA; Appendix A). However, the highest value was found in *C. nodosa* leaves from the mixed zones in site #2, and the lowest was found in *H. stipulacea* from monospecific meadow zone in site #1 (Tukey’s pairwise post-hoc test; Table 3). 

Elemental content values and patterns varied among plant parts of the two seagrasses. In leaves, both species showed a TC content generally higher in site #1 than in site #2 (two-way ANOVA; Appendix A). At both sites, *C. nodosa* showed the highest values in the monospecific zone, while *H. stipulacea*’s were in the mixed zone (Tukey’s pairwise post-hoc test; Table 3). In both species, the leaf δ^13^C (‰) content was coherently generally higher in site #1 than site #2 (two-way ANOVA; Appendix A). However, in both sites, *C. nodosa* leaf δ^13^C content was higher in the mixed than in the monospecific zones, while *H. stipulacea*’s leaf δ^13^C content was comparable between meadow zones (Tukey’s pairwise post-hoc test; Table 3). Conversely, the TN leaf content showed no differences between sites, species and meadow zones (two-way ANOVA; Appendix A). The leaf δ^15^Ν (‰) in *C. nodosa* it was higher in site #1 than in site #2, while in *H. stipulacea* it showed the opposite trend, though both species did not show significant difference between meadow composition in each site; thus, the differences between meadow zone were only due to the species (two-way ANOVA; Appendix A, Tukey’s pairwise post-hoc test; Table 3).

In rhizomes, one of the main carbon storage of seagrasses, the TC content was comparable between sites and generally higher in *C. nodosa* than in *H. stipulacea*, regardless of the meadow zones (two-way ANOVA; Appendix A). The rhizome δ^13^C (‰) content was generally higher in site #1 than site #2, as found in the leaves. In both sites, *C. nodosa* rhizome δ^13^C content was higher in the mixed than in the monospecific zones, while the opposite was found in *H. stipulacea* (Tukey’s pairwise post-hoc test; Table 3). Both the TN and δ^15^Ν (‰) content in rhizome showed the same trend without difference between species. Their content was higher in site #1 than site #2 in *C. nodosa*, while it was the opposite in *H. stipulacea* (two-way ANOVA; Appendix A, Tukey’s pairwise post-hoc test; Table 3). 

In roots, both the TC content and the δ^13^C (‰) followed the same pattern. They were generally higher in site #1 than site #2, and in *C. nodosa* than in *H. stipulacea* (two-way ANOVA; Appendix A, Tukey’s pairwise post-hoc test; Table 3). However, as for the other plant parts, roots δ^13^C content was higher in the mixed than in the monospecific zone in *C. nodosa* from both sites, while it showed the opposite trend in *H. stipulacea* (two-way ANOVA; Appendix A, Tukey’s pairwise post-hoc test; Table 3). 

The TN content in roots was comparable between sites, species and meadow zones, but the highest value was found in *H. stipulacea* from the mixed meadow zone of site #2 (two-way ANOVA; Appendix A, Tukey’s pairwise post-hoc test; Table 3). The δ^15^Ν (‰) content in roots, however, was higher in *C. nodosa* in the monospecific meadow zones compared to the mixed one, while it showed the opposite trend in both sites in *H. stipulacea*, showing the highest value in *H. stipulacea* from the mixed meadow zone from site #2. 

Results are reported in Table 3. 

### 2.3. Bacterial Communities

#### 2.3.1. Bacterial Community Diversity

The α-diversity of seawater microbial communities was higher in site #2 than in site #1, while the sediment bacterial communities’ α-diversity (Shannon Index, values in Appendix A) was highest in site #1.

In both sites, the sediment bacterial communities of the monospecific *C. nodosa* zones showed the highest α-diversity, presenting a significant positive relationship with seagrass density (R = 0.64, *p* < 0.05); conversely, the sediment α-diversity of monospecific *H. stipulacea* zones was the lowest, with no relationship with seagrass density.

The α-diversity of seagrass-associated bacterial communities was higher in site #1 than in site #2, showing a significant relationship with depth (R = −0.3; *p* < 0.05). Furthermore, the α-diversity was always higher in monospecific than in mixed zones (Shannon index diversity permutation test; *p* < 0.05), except for *H. stipulacea* from site #2. In both sites, the α-diversity was significantly higher in leaf-associated than in seawater bacterial communities (Shannon index diversity permutation test, *p* < 0.05), whilst the α-diversity was significantly lower in roots/rhizomes-associated than in sediment bacterial communities (Shannon index diversity permutation test; *p* < 0.05). The α-diversity of the bacterial communities associated with *C. nodosa* leaves showed a significant inverse relationship with total phenols (R = −0.63 *p* < 0.01); this was not found in *H. stipulacea*.

As expected, site or substrate were the main drivers shaping the bacterial communities (two-way ANOSIM, site R = 0.37, *p* < 0.05; sample type R = 0.40, *p* < 0.05). Thus, to remove major environmental variability and to detect the finest differences among the bacterial communities, subsets of microbial communities found in seawater were compared between sites; those found in sediment were compared between sites and meadow zones; those associated with each seagrass plant part were compared in each site between species and meadow zones. These analyses showed that seawater bacterial communities were similar between sites (ANOSIM, *p* = NS); conversely, the sediment bacterial communities were different between sites (two-way ANOSIM, site R = 0.70; *p* < 0.05) but not between meadow zones (two-way ANOSIM, meadow zone p = NS). Structure and composition of the seagrass-associated bacterial communities were driven by seagrass species and not by meadow zone. A significant relationship with the type of meadow was only found in site #2 for the leaf-associated bacterial communities (ANOSIM statistics in Table 4; Figure 1a–d).

#### 2.3.2. Environmental Bacterial Communities’ Composition and Indicator Species

The most abundant bacterial group found in seawater in both sites was SAR11_clade order (Alphaproteobacteria class, representing 52% and 53% of the entire community, respectively, in site #1 and #2). 

In the shallow site (#1), the Indicator Species Index (IndVal) indicated that Flavobacteriaceae, Pseudoalteromonadaceae (Gammaproteobacteria), and Lentimicrobiaceae (Bacteroidetes) were significantly (*p* < 0.05) associated with monospecific *C. nodosa* colonized sediment, while Fusobacteriaceae (Bacteroidetes) and Izemoplasmataceae (Bacilli) were significantly (*p* < 0.05) associated with mixed meadow sediment. Conversely, any bacterial taxon was found significantly associated with *H. stipulacea* colonized sediment. In the deep site (#2), the IndVal indicated that Vibrionaceae, Alteromonadaceae (Gammaproteobacteria), and Thermoanaerobaculaceae (Thermoanaerobaculia) were significantly associated with *C. nodosa*-colonized sediment, while Izemoplasmataceae were significantly associated with mixed meadow sediment; likewise, in the shallow site, no bacterial taxa were specifically associated with *H. stipulacea*-colonized sediment (Table 5, which also includes the putative bacterial metabolic traits and functions).

#### 2.3.3. Composition of Seagrass Core Communities

The amplicon variant sequence (AVS, i.e., each inferred single DNA sequences recovered from a high-throughput analysis of 16S rDNA genes, differing at single-nucleotide level) distribution highlighted that shallow site plants harboured a higher number of AVS than deep site plants (2624 vs. 2189 AVS), and that, in both sites, the belowground bacterial communities included a higher number of AVS than the aboveground (site#1: 1146 vs. 1478 AVS; site #2: 1048 vs. 1141 AVS). In detail, Venn diagrams revealed that the bacterial communities associated with both seagrasses shared some AVS which constitute the above/belowground bacterial core (Figure 2, central sections), but each sample also hosted unique AVS (Figure 2, marginal sections). In both seagrasses from monospecific or mixed meadows, the bacterial cores were composed of a low number of AVS (aboveground 90 vs. 33, belowground 52 vs. 23, in site #1 and #2, respectively), represented by a high number of reads (aboveground 50.1% vs. 34%, belowground 71.6% vs. 41%, in sites #1 and #2, respectively).

Taxonomically, the Comamonadaceae and Rhodobacteraceae were widespread and abundant in all seagrass samples. Moreover, other bacterial taxa were found associated with both the above and the belowground parts of seagrass from both sites, hence ubiquitous taxa.

The aboveground bacterial core included five bacterial families and one still unknown bacterial group: Comamonadaceae (Gammaproteobacteria), Rhodobacteraceae (Alphaproteobacteria), Flavobacteraceae (Flavobacteriia), Rhizobiaceae (Alphaproteobacteria) were the most abundant. At percentages lower than 10%, Rhodocyclaceae (Betaproteobacteria), Nodosilinaceae (Cyanobacteria), and an uncultured family of Alphaproteobacteria were also found. 

The belowground bacterial core included eleven bacterial families; some of them, such as Comamonadaceae, Rhodobacteraceae, Flavobacteraceae, and Rhizobiaceae (Alphaproteobacteria), were shared with the aboveground bacterial core and were the most abundant across all sample types. Other less represented families were Saccharospirillaceae (Gammaproteobacteria), abundant mainly in the *C. nodosa* belowground bacterial community from the shallow site, and Stappiaceae (Alphaproteobacteria), abundant mainly in the *H. stipulacea* belowground bacterial community from the deep site.

#### 2.3.4. Host-Specific Bacterial Associations and Indicator Species

The great majority of amplicon variant sequences (AVS) were associated only with one of the two seagrasses, i.e., they were unique. In both sites, *C. nodosa* and *H. stipulacea* plants in the monospecific zones hosted the highest number of unique AVS in the aboveground communities (Figure 2a,b). Similarly, belowground communities associated with *C. nodosa* hosted the highest number of unique AVS in monospecific stands (Figure 2c), while *H. stipulacea*-associated belowground communities hosted the highest number of unique AVS in mixed stands (Figure 2c,d). Furthermore, in both sites, each seagrass from the mixed zones shared more AVS with its counterpart from the monospecific zone than with the other syntopic species. 

In the shallow site, the IndVal highlighted that different bacterial taxa were significantly associated with each plant part of each seagrass, in each meadow zone (*p* < 0.05; Figure 2). The number of bacterial taxa significantly associated (in red) with aboveground bacterial communities was six in *C. nodosa* monospecific and four in mixed stand, while it was three in *H. stipulacea* monospecific and only one in the mixed stand. The number of bacterial taxa significantly associated with belowground communities was three in *C. nodosa* monospecific and five in the mixed stand, while only one was associated with *H. stipulacea* monospecific and none with the mixed stand. Still in the deep site, the IndVal highlighted a significant association of different bacterial taxa with each plant part of each seagrass, in each meadow zone (*p* < 0.05; Figure 3). The number of taxa significantly associated with the aboveground communities were four in *C. nodosa* monospecific and only two in the mixed stand, while five were associated with *H. stipulacea* monospecific and seven with the mixed stand. The number of taxa significantly associated with the belowground communities was ten in the *C. nodosa* monospecific and only three in the mixed stand, while six were associated with *H. stipulacea* monospecific and nine with the mixed stand. The host-specific bacterial taxa significantly associated with aboveground or belowground plant parts in each site are reported in Table 6, along with their putative metabolic traits or functions.

## 3. Discussion

The current study aimed to evaluate the differences in plant and bacterial descriptors of the Mediterranean native seagrass *Cymodocea nodosa* and the exotic seagrass *Halophila stipulacea* growing in a specific spatial distribution found at two different sites and depths in the Aegean Sea (see Figure 4): monospecific zones of each species converged into a mixed one, where plants grew in syntopy (*sensu* Rivas [121]). This spatial distribution allowed us to consider the plants of each species in the monospecific zone as a ‘control group’ to be compared with those from the mixed zone as an ‘experimental group’, in a sort of natural experiment. 

The comparison of these ‘groups’ allows one to highlight the holobiont changes by analysing the variation of seagrass descriptors (morphometric and biochemical) and associated bacterial community (structure and composition) under interspecific interactions. 

Plant descriptors and bacterial community structure and composition indicate that both descriptors differed between sites because of the abiotic factors and, in both sites, they changed in each species between monospecific and mixed meadows due to interspecific interaction. Plant descriptors indicated a stress condition due to the competitive interaction, and the epiphytic bacterial communities varied with the stressed condition. The two seagrasses responded differently to syntopy. Namely, the native *C. nodosa* showed reduced density, significantly higher phenol content and a reduction in the associated bacteria. This reduction occurred in mixed zone where *C. nodosa* grew intermingled with *H. stipulacea*, concurrently with morphometric and biochemical variations, and it may represent a further indication of stress. Conversely, the exotic *H. stipulacea* did not show reduced density in the mixed meadows; notwithstanding, the high phenol content and the associated microbes remained almost unaffected.

These results are further integrated by the interesting effect of monospecific and mixed meadows on the diversity of sediment bacterial communities: the two seagrasses differently influenced the sediment communities, despite the same abiotic characteristics of the sediment. In monospecific *C. nodosa* zones, the diversity of the sediment bacterial community was the highest, being directly related with seagrass density. In fact, the long and thick *C. nodosa* roots were able to enrich the sediment with oxygen, glucose, and carbohydrates [122], increasing the chemical micro-niches available for microbes [19,116,123]. Conversely, under monospecific *H. stipulacea*, the sediment bacterial diversity was the lowest because the thin and short roots of *H. stipulacea* did not support the bacterial diversity, being less effective in changing sediment conditions [30,42,124]. The different contribution of each seagrass to the sediment bacterial communities was clear in mixed zones, where sediment bacterial diversity had intermediate values. Insights into the holobionts’ competition dynamic in both the plant and the bacterial partners are reported below, including the role of the bacterial core, which supports the basic plants’ functions, and the host-specific bacterial components, which highlight different seagrass–microbiota relationships under competition.

### 3.1. The Plant Partner of the Holobiont Highlights Competition Dynamic

While in syntopy, organisms (particularly the sessile ones, such as plants) must partition the available resources, resulting in a competitive interaction which reduces resource availability to both partners, usually in an unsymmetrical way [125]. According to this statement, seagrasses in the mixed meadow zones were found to rearrange their density, morphometry, and physiology, due to their high level of interaction.

The most evident unsymmetrical trait of this dynamic was density: in both sites, *C. nodosa* clearly reduced its density in response to *H. stipulacea*’s tight proximity; in contrast, the exotic seagrass density remained unchanged. Another unsymmetrical trait, due to the foreseeable resource partitioning, was the leaf area reduction in mixed meadow zones found in both species, though significant only in *C. nodosa*. These data suggested that *H. stipulacea* may successfully compete with *C. nodosa*, as already suggested by previous studies based on density variation [126]. 

Biochemical descriptors and seagrass elemental composition provided further insights into the competitive interaction dynamics. Pigment contents did not show remarkable differences between the two species in both meadow zones, including a significant increase in Chl *b* in the plants of both species from the mixed meadow; this increase was particularly evident in *H. stipulacea* from site #1, probably due to *C. nodosa* shadowing [40]. Total phenol content increased in both species and in both sites in mixed meadows, with the highest value found in *C. nodosa* from the mixed meadow zone of site #2. Total phenol content increase is considered a stress-induced defence mechanism and a reliable seagrass descriptor for several kind of disturbances [127,128,129,130,131,132,133,134]. Total phenols’ increase has been already reported in the case of competition between *P. oceanica* and the invasive seaweed *Caulerpa taxifolia* [132,133,134]. To the best of our knowledge, the total phenols increase in seagrass–seagrass interaction was never assessed before, though our results suggest that this increase can be considered a response to interspecific competition, at least for *C. nodosa*. 

The values and the differences of carbon content between seagrasses found in this study were within the natural range and the difference depends on interspecific variation [135,136]; this indicated that plants maintain their physiological features, in spite of different sediment carbon content. Conversely, the isotopic content indicated different growth dynamics of the two species and gave an insight into their dynamics under competition. In fact, in both sites and meadow zones, *C. nodosa* always contained a higher concentration of δ^13^C‰ than *H. stipulacea*. These differences may depend on a higher efficiency of *H. stipulacea* in absorbing light carbon isotopes due a fast growth rate (0.2 cm d^−1^, [36] for *Halophila* vs. 0.004 day^−1^ for *Cymodocea*; [136]), which may represent a potential advantage of the exotic over the native seagrass, especially in nutrient-limiting conditions [64]. 

A different dynamic has been found for nitrogen: in the sediment of the two sites, total nitrogen did not show differences, though it changed in plants’ tissues. This is probably due to the associated N-fixing bacteria, which is able to supply plants with the vital nutrient [13]. However, in the mixed zone, the δ^15^N content was higher in *H. stipulacea* than in *C. nodosa* leaves; this suggests that the exotic species must be able to absorb nitrogen from other matrices, i.e., from a water column, in a more efficient way than the native one.

The overall better performance of *H. stipulacea* seems to be counterintuitive, as the high canopy of *C. nodosa* may shade light to *H. stipulacea* while in syntopy. Nevertheless, *H. stipulacea*’s high plasticity and growth rate means the plants are able to escape this risk: the leaves become larger under lower light conditions and its fast growth rate makes this species a good competitor even against taller plants. These results suggest that, due to the different growth rate and structure of the two seagrasses, the main resource they compete for is space, which obviously implies the availability of many resources. *H. stipulacea*, due to a faster growth rate, can easily ‘occupy’ space, hindering the growth of the native species.

### 3.2. The Bacterial Partner of the Holobiont

The epiphytic bacterial communities’ structure and composition associated with seagrasses are known to be composed of two distinguishable components. The ‘bacterial core’ is an ubiquitous component, common to both species and present in all plant parts and sites; therefore, it is probably involved in constitutive seagrass processes [137]. This component colonizes the seagrass surfaces due to their wide distribution and the surface-adapted lifestyle. Nevertheless, the entity of colonization, higher on the seagrass than in the environmental pool, suggests that they find suitable micro-niches on seagrass, as suggested by the literature data [17,18,19,22,69,137]. However, it cannot be excluded that some bacterial groups may be opportunistic colonizers, too. The ‘host-specific component’ is a species-specific component, associated with each seagrass species in each meadow zone. This component has been related to the host physiology, due to its variable occurrence in different conditions [137]. The Indicator Species Index evaluated the statistical significance of the host-specific associations [138]. 

#### 3.2.1. The Bacterial Core May Support the Basic Plants’ Functions

The most abundant families associated with seagrass tissues of both species were Comamonadaceae and Rhodobacteraceae, fostered by their wide environmental distribution and by their surface-adapted lifestyle [35]. Comamonadaceae includes photoautotrophic, photoheterotrophic bacteria, anaerobic fermentative bacteria, anaerobic denitrifying and iron-reducing bacteria [139]. When found in the rhizosphere of terrestrial plants, these are considered growth-promoting and biocontrol agents [139,140,141]. Rhodobacteraceae includes biofilm forming bacteria [96,97,142], aerobic photo- and chemo-heterotrophs, adapted to symbiosis with aquatic organisms [96]. This family is involved in both sulphides’ oxidation, keeping their concentration at a seagrass-tolerable concentration [86], and in antibacterial compounds’ production, a deterrent for other bacteria [116,140]. Other core component families are Rhizobiaceae and Rhodocyclaceae, involved in nitrogen-fixing [61,122], and able to use a wide range of carbon sources and electron acceptors [143]. Flavobacteraceae operate extensive degradative activity [87], providing a nutrient-rich environment. This family has been found in the rhizosphere of successfully restored seagrasses [115] and are associated with plant-growth-promoting traits [144]. Furthermore, the belowground bacterial core includes Rhodobacteraceae-related families, involved in nitrification (Stappiaceae [94,145]) and denitrification processes (Burkholedriaceae [146,147], Caulobacteraceae, [24,118,122], Saccharospirillaceae, and Oceanospirillales, [85,86,117,148,149]), some of which were already found to be associated with seagrass tissues [61,124]. The putative metabolic traits of the dominant bacterial groups hint at the involvement of the bacterial core in basic processes of the host plant as, among others, nutrient cycling and epiphytes growth control.

#### 3.2.2. Host-Specific Bacterial Component Highlights Different Seagrass–Microbiota Relationship under Competition

According to their diversity and composition, the host specific components of the seagrass-associated bacterial communities seemed to be more tightly linked to the ‘ecophysiological’ conditions of the plants. 

The leaf-associated bacterial communities showed higher diversity in *C. nodosa* than in *H. stipulacea*. This probably depended on the leaves’ structure and life span: the long and persistent *C. nodosa* leaves allow the settlement of a more structured biofilm than the tiny and short-lived leaves of *H. stipulacea* [30,150]. This dynamic has already been suggested by comparing the leaf colonizers of *P. oceanica* and *H. stipulacea* [22]. 

In both species, on the leaves, the host-specific bacterial diversity was lower in mixed than in their monospecific meadows, except for *H. stipulacea* in the deep site. *C. nodosa* leaves harboured a rich set of taxonomically diversified microbes in almost all conditions, except in the mixed zone of the deep site (#2), where plants were clearly affected by syntopy, as suggested by the suite of descriptors, and the associated bacteria seemed to be affected as well, as the set of host-specific microbes was narrow. In the stressed condition, the reduction in host-specific bacteria on *C. nodosa* leaves was significantly related with the increased phenol content due to syntopy, which may inhibit the bacterial association [26,28,131]. Conversely, in *H. stipulacea* the host-specific bacteria associated with leaves did not show a clear pattern either in the community structure or in the taxonomic composition across all conditions. This high variability of the associations could be linked to the well-known plasticity of the exotic seagrass. 

On roots/rhizomes, the *C. nodosa* host-specific associated bacteria were numerous and diverse in all the conditions, as on leaves, except for the mixed meadow zone from the deep site (#2), where the host-specific belowground bacterial associations were narrow. This suggested a reduced capability of the host to sustain the rich bacterial community, due to the reduced plant density and/or roots’ activity [16,17,18,19]. Mirroring the aboveground results again, the *H. stipulacea* belowground host-specific bacterial associations showed high variability, ranging from no significant association in the mixed zone of the shallow site (#1) to a rich and wide set of associated bacteria in both meadow zones of the deep site (#2). This last result further accounts for *H. stipulacea*’s capability to associate with epiphytic bacteria in different ways, probably depending on its high morpho-physiological plasticity [26,28].

Overall, the results showed that *C. nodosa*-associated bacterial communities were altered in plants from mixed zones (hence in synthopy), concurrently with altered biometric and biochemical descriptors, highlighting a plant stressed condition. This dynamic was particularly evident in the deep site (#2), where more severe signs of stress were reported in *C. nodosa* due to the proximity of *H. stipulacea*. Conversely, *H. stipulacea* was shown to be able to thrive well even when associated with different types and assemblages of host-specific bacteria, as confirmed by the biometric and biochemical plant descriptors.

This response suggests a further «generalist attitude» of *H. stipulacea*, i.e., the capability to associate with whatever available microbe, which may be another key component of its adaptive capability.

## 4. Materials and Methods

### 4.1. Sampling Design and Site Description

In this study, both the seagrass host features and the epiphytic bacterial communities of the native *C. nodosa* and the exotic *H. stipulacea* have been investigated in a specific condition. The two seagrasses formed a single meadow in which three different contiguous zones can be distinguished: a monospecific zone of *H. stipulacea*, a mixed one, and a monospecific zone of *C. nodosa* (Figure 4). This condition was found in two sites located at different depths along the northern coast of Crete (Greece). In both sites, the mixed zones were smaller than the monospecific ones. In the shallow site, the mixed meadow covered an area of approximately 30–40 m^2^, while the extension was of about 20–30 m^2^ in the deep site.

Data from the monospecific zone of each seagrass species, considered a control dataset, were compared with those collected in the mixed zone. This topology has been considered a ‘natural experiment’ of seagrass interaction, aimed at evaluating the seagrass holobiont response under putative competition, at two different depths. The seagrass host characteristics were evaluated using standard descriptors (meadow density, leaf area, rhizome length, or plant pigment, total phenol, elemental content, and isotopic composition). The bacterial communities’ structure and composition associated with the above- or belowground seagrass tissues were evaluated using the 16S rRNA amplicon sequencing.

The study was carried out at the beginning and at the end of summer 2019, with the temperature ranging from 26 to 24.7 °C (EMODnet), a period in which both species are considered to thrive well [34,45], in two sites in the northern coast of Crete (Greece; Figure 5a,b). The first site (called #1-shallow) was off the Kalami beach, on the southern coast of Souda Bay (35°28′08″ N 24°08′58″ E); it is a semi-closed bay with sandy sediments and a gentle slope. The meadow was found at 7 m depth. The second site (called #2-deep) was placed off the shore of Mononaftis Bay (35°24′59″ N, 25°01′12″ E), where the sediment was coarse and the bottom slope steeper. The meadow was found at 14 m.

Sampling was conducted by SCUBA diving in each meadow zone (monospecific *C. nodosa*, monospecific *H. stipulacea* and mixed *C. nodosa*–*H. stipulacea* stands) of both sites, as reported in Figure 5c. Samples of seawater, pore water, sediment and plants were collected as reported below and immediately transported to the lab, kept cold under dark conditions until further processing.

### 4.2. Abiotic Settings

One replicate per site of bottom seawater (200 mL each) was collected right above the plants’ canopy using a syringe. In addition, triplicates of pore water samples per site and meadow zone (10 mL each) were collected in random chosen spots using a perforated tube that penetrated the sediment down to 10 cm. Samples were immediately filtered through 0.45 μm membrane filters (Millipore) into polyethylene bottles and stored at −20 °C until further analysis. Samples were analysed for phosphate (PO_4_^3−^), according to Strickland and Parsons [151] (1972; detection limit 0.05–4 μM), and ammonium (NH_4_^+^), according to Ivancic and Degobbis [152] (1984; detection limit 0.1–10 μM). Triplicate sediment samples were collected in each site and meadow zone using a core (internal diameter, 3 cm) down to 10 cm sediment depth. Sediment samples were dried at 60 °C for 72 h, ground and processed according to Vizzini and Mazzola [153]. Each sample was divided into two aliquots of around 30 mg: one for total carbon (C) and nitrogen (N) content and δ^15^N analysis, the other for δ^13^C analysis. Before δ^13^C analysis, samples were acidified with drop-by-drop 1N HCl to remove traces of carbonates and subsequently dried and powdered again. All samples were weighed in tin capsules and analysed through an elemental analyser-isotope ratio mass spectrometer system (Thermo Flash EA 1112—IRMS Delta Plus XP). Isotopic data were reported in common delta units (δ) and referred to VPDB and atmospheric nitrogen standards for carbon and nitrogen, respectively, following the formula: $$\delta^{x}=\left[\left(\frac{R\;sample}{R\;standard}\right)-1\right]*10^{3}\;$$
where x is the stable isotope mass (13 for C, 15 for N), and R is the corresponding ^13^C/^12^C or ^15^N/^14^N ratio. Analytical precision based on the standard deviation of replicates of internal standards (International Atomic Energy Agency IAEA-NO^−^_3_ for δ^15^N and IAEA-CH-6 for δ^13^C) was 0.2‰ for both δ^13^C and δ^15^N.

For each site, triplicates of 20 g, one core per meadow zone, of dried sediment samples were used to analyse the granulometry by shaking samples for 10 min with sieves equipped with 1000, 250, 125 and 63 μm meshes.

### 4.3. Biotic Settings

Seagrass shoots were collected (five replicates) in randomly chosen spots, distanced at least five meters in each meadow zone within each site (Figure 4c), to analyse structural and biochemical variables using aluminium corers (internal diameter 15 cm). For the analyses, 50 mg of fresh weight leaves per replicate were used, i.e., 3–4 cm of a 2nd leaf for *C. nodosa* and 1–3 leaves from a single fragment for *H. stipulacea*. To evaluate the meadow density, all the plants within each corer were rinsed and counted (from 50 to 194 shoots for *C. nodosa,* and from 309 to 3772 shoots for *H. stipulacea*). For morphometrical analyses, all rinsed plants were digitally photographed to quantify the leaf area (from 15 to 90 leaves per replicate, depending on the shoots’ density; data were normalized) and the total length of rhizomes per square meter (TLR, total length of rhizomes, including branches, present in a 25 × 25 cm quadrat and then normalized to square meter) using Image J software 1.53e [154]. For elemental and isotopic quantification, all the collected shoots were separated into leaves, rhizomes and roots, epiphytes were gently removed, and seagrass tissues were dried (60 °C for 72 h), treated and analysed as described above for sediment. For pigment and total phenol content, five 2nd leaves per meadow zone were gently cleaned from the epiphytes and grounded in liquid N_2_ using mortar and pestle, following the protocol by Migliore et al. [127]. Chlorophyll-*a*, Chlorophyll-*b* and total Carotenoids were extracted from 50 mg of fresh weighted leaves and quantified using the formulas of Lichtenthaler and Buschmann [155]; data were expressed as mg g^−1^ fresh weight:$$\mathrm{Chl}\;a=16.72(A{\;}_{665}-A_{750})-9.16\;\left(A_{652}-A_{750}\right)$$
$$\mathrm{Chl}\;b=34.09(A{\;}_{646.8}-A_{750})-15.28\;\left(A_{663.8}-A_{750}\right)$$
$$\mathrm{Car}=[(1000*(A{\;}_{470}-A_{750})-\left(1.63*\mathrm{Chl}\;a\right)*\left(104.96*\mathrm{Chl}\;b\right)]/221$$
where Chl *a* = Chlorophyll *a*; Chl *b* = Chlorophyll *b*; Car = Total Carotenoids. 

Total phenols were extracted from 50 mg of fresh weighted leaves and quantified, expressed as chlorogenic acid equivalents (mg) per gram of plant material (fresh weight), using the formula below. A calibration curve was made with chlorogenic acid using five different concentrations (0, 25, 50, 100 and 200 µg ml^−1^).
$$\mathrm{Total}\;\mathrm{phenols}=\left(A_{724}-0.0821\right)/r)$$
where *r* is the calibration curve coefficient from the chlorogenic acid measurements.

### 4.4. Bacterial Community Analysis

In each site and meadow zone, bacterial communities were collected separately in triplicates from above- and belowground seagrass tissues and from sediment and seawater. Due to the different plant morphology, each aboveground replicate consisted of a 2nd leaf of *C. nodosa* or twelve leaves of *H. stipulacea*. Each belowground replicate consisted of 6 cm of rhizome/roots from a *C. nodosa* shoot or about 10 cm of rhizomes/roots (supporting around 6 shoots) from a *H. stipulacea* fragment. 

The tissues were carefully rinsed with 2 mL of a washing solution (200 mM Tris–HCl pH 8, 10 mM EDTA, and 0.24% Triton X-100; [156]; the solution was centrifuged (20′, 5000× *g*), and the pellet stored in 2 mL of a transport solution (Tris 10 mM, EDTA 50 mM; [156], as reported by Mejia et al. [26].

Triplicate of bottom seawater samples (1 L per replicate) were collected underwater and filtered using a sterile 0.2 μm Whatman^®^ membrane filter. The filters were stored submerged in transport solution until DNA extraction. Triplicate sediment samples were collected using mini corers (internal diameter 1.5 cm). The sediment was stirred and 2 g of mixed sediment per replicate was stored submerged in transport solution until DNA extraction.

The bacterial metagenomic DNA was extracted using the Power Soil^®^ DNA isolation kit (Mo Bio, Carlsbad, CA, USA) according to the manufacturer’s instructions. The 16S RNA gene has been amplified using PCR with the universal primers Com1 (forward, 5′-AGCAGCCGCGGTAATAC-3′) and Com2 (reverse, 5′-CGTCAATTCCTTTGAGTTT-3′); the amplified DNA was then purified using DNA soil extraction kit (GeneAid, Taiwan). The pure DNA extracts (≥5 ng μL^−1^) were sent to Molecular Research LP (MR DNA Shallowater, TX, USA) for NGS Illumina MiSeq sequencing platform. The raw paired-end sequences obtained were analysed using Quantitative Insights Into Bacterial Ecology (QIIME 2.10; [157]. The sequences were demultiplexed, quality- and chimera-checked, and filtered using the DADA2 plugin [158]. In total, 5419 AVS (amplicon variant sequence) were identified, with a total frequency of 1,156,166 reads. Taxonomic identification of the 16S rRNA gene sequences was performed using a Naive Bayes classifier trained with the SILVA 138 SSU database [159]. AVS classified as chloroplasts or mitochondria were discarded from the dataset. The rarefaction curves, built to evaluate differences and efficiency in the sampling effort, confirmed that the sequencing coverage was good (Appendix A). The dataset was normalized at the common depth of 1553 sequences per sample, the lowest number of sequences in the dataset (sample *H. stipulacea*, belowground, monospecific meadow, site #2, replicate #1), to keep three biological replicates of each sample type in the dataset. The final dataset was composed of 5419 AVS. This targeted locus study project has been deposited at GenBank PRJNA826444 under the accession KFUQ0100000. 

### 4.5. Data Analysis

To test whether each inorganic nutrient differed in content between sites in bottom water, a *t*-test was used (two levels: site #1 and #2). 

Two-way ANOVA (PAST 4.05, [160]) was used to test whether significant differences existed in porewater inorganic nutrient content between sites (two levels: site #1 and #2), and among meadow zones (four levels monospecific *H. stipulacea*, monospecific *C. nodosa*, mixed meadow *H. stipulacea* and mixed meadow *C. nodosa*), using the site and the meadow zone as fixed factors. Two-way ANOVA was also used to test whether significant differences existed in the sediment isotopic content using site, depth, (two levels: site #1 and #2), and meadow zone (three levels: *H. stipulacea* monospecific, *C. nodosa* monospecific, mixed stands). Differences in sediment granulometry between sites were tested using PERMANOVA and expressed as a percentage of each granulometry range per gram of sediment.

Two-way ANOVA (PAST 4.05, [160]) was used to test whether significant differences existed among each seagrass morphological descriptors (seagrass morphometry, density) using the site (two levels: site #1 and #2) and the meadow zones for each species (two levels each monospecific and mixed zones), as the biometry of the two species cannot be compared. For biochemical descriptors (pigments’ content, total phenol and the isotopic content of each seagrass tissues type, leaves, roots, rhizomes), two-way ANOVA was applied using the site (two levels: site #1 and #2) and the type of samples, whereas the species in each meadow composition (four levels: monospecific *H. stipulacea*, monospecific *C. nodosa*, mixed meadow *H. stipulacea* and mixed meadow *C. nodosa*) were used as fixed factors in order to compare each seagrass species in each condition.

Prior to ANOVA, the Shapiro–Wilk test was used to check the normality of the data and the Levene’s test for homogeneity of variance; when necessary, the log_10_ data transformation was applied. In the case of significant differences (*p* < 0.05), Tukey’s pairwise post-hoc test was used to show the significant interaction between factors. The statistical test was performed using PAST 4.05.

All the statistical analyses were performed using the normalized AVS dataset, which indicated slight differences among the different DNA samples collected in the field at a finer scale than family. Conversely, the dataset agglomerated at the family level was used to visualise and describe the bacterial community differences among samples. QIIME2 [157] or PAST 4.05 [160] software was used.

Bacterial diversity within the sample (α-diversity) was estimated using the Shannon index (PAST 4.05). Differences in α-diversity were assessed with the permutation diversity test (PAST 4.05) between bacterial communities associated with the same species and part of each seagrass, using the meadow zone as a variance factor. Kruskal–Wallis comparison was used to compare the diversity of sediment bacterial community of each meadow zone and site. Pearson correlation was used (QIIME2, alpha correlation plugin) to test the possible relationship between aboveground bacterial α-diversity and total phenols or sediment bacterial α-diversity and seagrass density. 

Two-way ANOSIM was used to evaluate significant differences of β-diversity in all of the dataset samples, using the site and the sample type, as well as the substrate in which bacterial communities grow. Thus, to detect finer differences, subsets of sample types were compared. ANOSIM was used to compare seawater microbial communities between sites. Two-way ANOSIM was used to compare sediment microbial communities between sites and meadow zones, and to compare bacterial communities associated with seagrass in each site, using the species and meadow zone as source of variance. These latter were visualised with PCoA (PAST 4.05). 

Venn diagrams were built to visualise unique or shared AVS among seagrasses’ parts and meadow zone (https://bioinformatics.psb.ugent.be/webtools/Venn/ (accessed on 1 December 2021). Taxa agglomerated at the family or lower level with a frequency above 50 reads were used to describe: (i) the above/belowground bacterial core, defined as the microbes associated with seagrass above- or belowground parts, despite the site and host species; and (ii) the host-specific bacterial associations and the taxa characterizing the sediment of each meadow zone, evaluated with the Indicator Species Index (henceforth referred to as IndVal; [138] Dufrene and Legendre, 1997), defined as taxa preferentially associated with a sample type such as the following: $$IndVal\;\left(\%\right)=100*\left(A_{ij\;}*\;B_{ij}\right)$$
$$A_{ij\;}=N_{ij}/N_{i}$$
$$B_{ij}=Nsites_{ij\;}*\;Nsites_{j}$$
where *A* is the specificity and *N_ij_* is the mean number of reads of species *I* across sites (samples) in group *j* (replicates of the same sample type were pooled), and *N_i_* is the sum of the mean numbers of individuals of species *i* over all groups; *B* is the fidelity and *Nsites_ij_* is the number of sites (samples) in group *j* (replicates of the same sample type were pooled) where species *i* is present, and *Nsites_j_* is the total number of sites in group *j*. The statistical significance of the indicator was estimated using random reassignments (9999 permutations) of sites across groups ([160] PAST 4.05, Hammer et al., 2001) and visualised as a heatmap ([161] Rstudio, pheatmap package).

## 5. Conclusions

This study, coupling seagrass descriptors with bacterial analysis, provided insight into the interaction between the exotic *H. stipulacea* and the native *C. nodosa* within the holobiont frame, while growing intermingled in two sites of the Aegean Sea. A clear stressed condition (and a possible decline) of *C. nodosa* seems to be the output of the seagrass synthopy, due to the putative competition with *H. stipulacea*; conversely, almost no effects were found in *H. stipulacea*.

These results highlighted the potential capability of *H. stipulacea* to outcompete the native *C. nodosa* in the Aegean Sea, relying on its morphophysiological plasticity and on the ability to harbour and interact with diversified bacterial communities. 

This work underlined the importance of investigating the host–microbiota interactions to unveil complex ecological processes, such as the competition, and again suggest the possible use of bacterial communities as a putative seagrass descriptor, although further studies are needed to foster their use in seagrass monitoring and conservation. 

## Figures and Tables

**Figure 1 plants-12-00350-f001:**
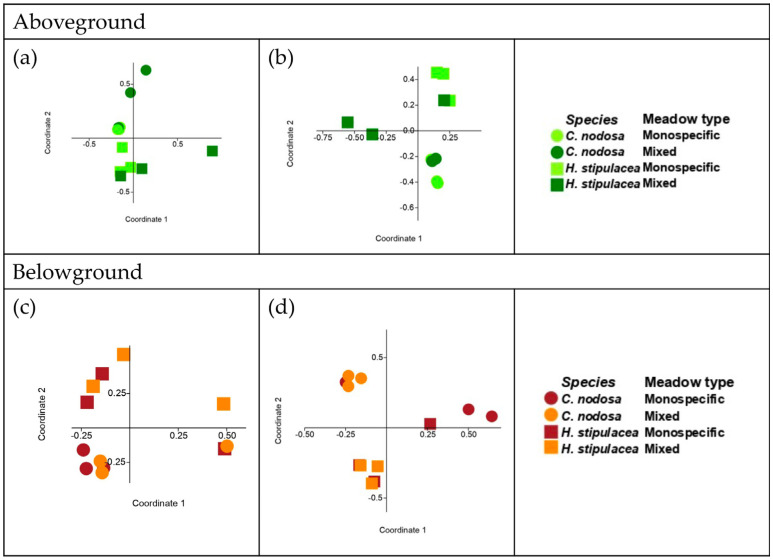
PCoA analysis with Bray-Curtis dissimilarity matrix of the aboveground-associated bacterial communities in site #1 (**a**) and #2 (**b**) and of the belowground-associated bacterial communities in site #1 (**c**) and #2 (**d**).

**Figure 2 plants-12-00350-f002:**
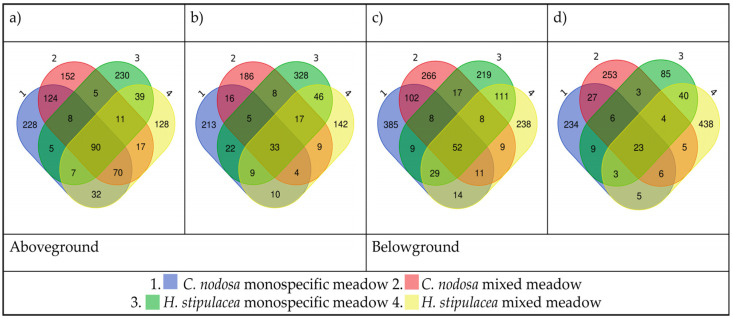
Venn diagram of the amplicon variant sequence (AVS) distribution on the aboveground seagrass tissues from the shallow site (#1; (**a**)), and the deep site (#2; (**b**)), or on the belowground seagrass tissues from site #1 (**c**) and site #2 (**d**).

**Figure 3 plants-12-00350-f003:**
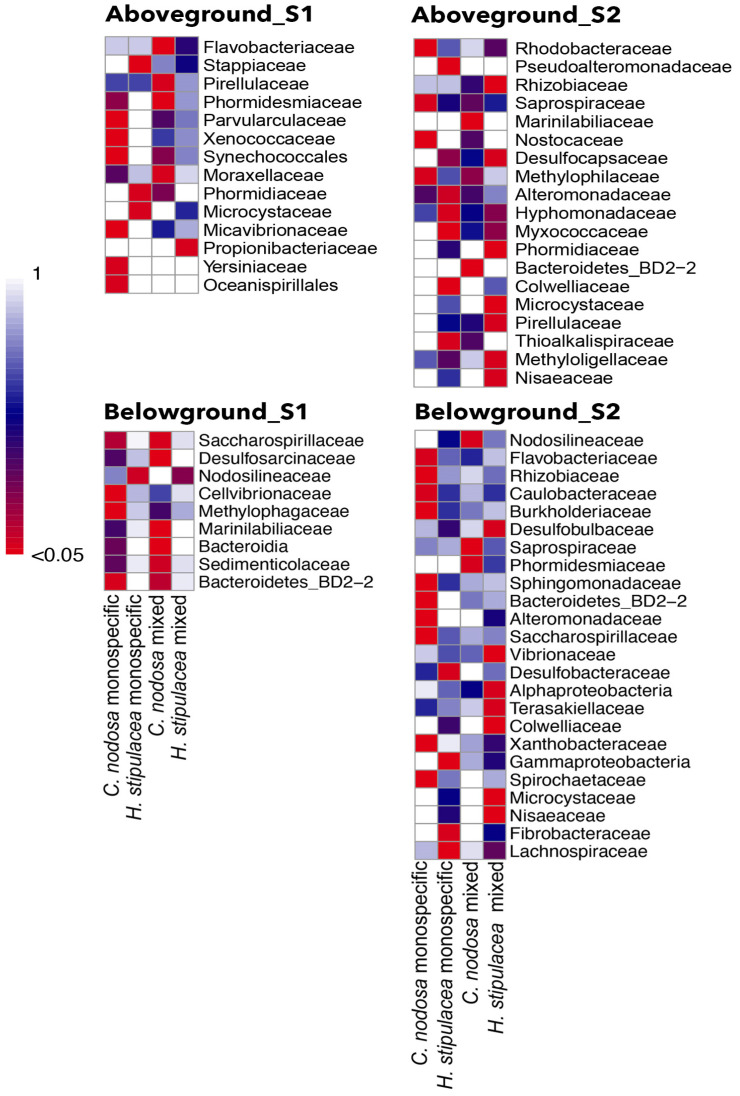
Heatmap visualization of the bacterial taxa significantly associated with the aboveground or belowground plant part using the Indicator Species Index (IndVal) in each site; significant *p*-value (<0.05) is reported in red.

**Figure 4 plants-12-00350-f004:**
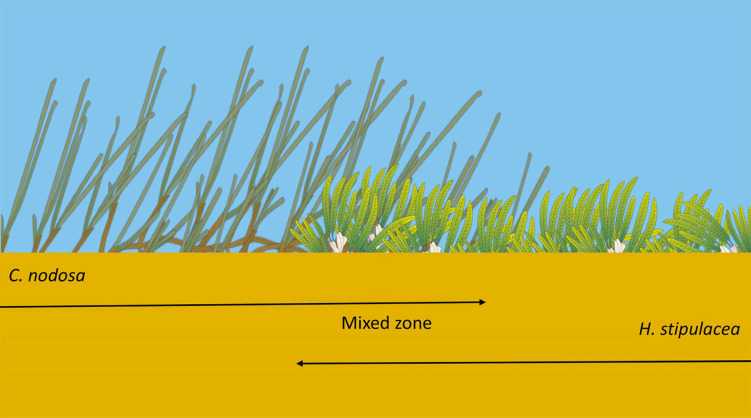
Seagrass pattern distribution in each sampling site: in the same meadow, a monospecific *C. nodosa* zone, a mixed zone, and a monospecific *H. stipulacea* zone can be found in a 50 m transect.

**Figure 5 plants-12-00350-f005:**
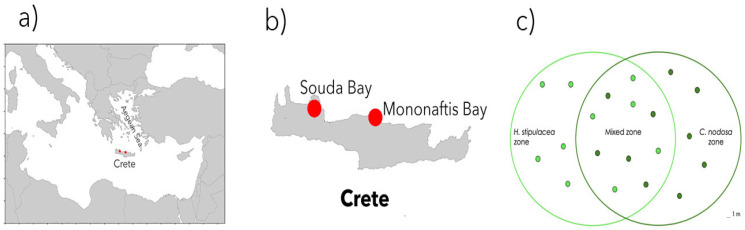
Sampling in the Crete Island (**a**); the sampling sites, Souda Bay and Mononaftis Bay (**b**); the sampling design, with randomly chosen sampling points (**c**).

**Table 1 plants-12-00350-t001:** Characterization of bottom seawater (mean values ± standard deviation). Significant differences between sites (*t*-Student test) are indicated by capital letters.

Bottom Water	Site #1	Site #2
NH_4_^+^ (µM)	0.39 ± 0.16	0.6 ± 0.35
NO_3_^−^ (µM)	0.46 ± 0.17 A	0.14 ± 0.00 B
NO_2_ (µM)	0.00 ± 0.00	0.00 ± 0.00
PO_4_^3−^ (µM)	0.06 ± 0.01	0.03 ± 0.01

**Table 2 plants-12-00350-t002:** Characterization of abiotic matrices (mean values ± standard deviation). Significant differences between the sites (two-way ANOVA Tukey’s post-hoc test) are indicated by capital letters, and in lowercase between meadow zones.

Site		Site #1						Site #2		
Species		*C. nodosa*		*H. stipulacea*		Mixed		*C. nodosa*	*H. stipulacea*	Mixed
Pore water	NH_4_^+^(µM)	5.7 ± 3.2		4.8 ± 2.2		2.3 ± 0.3		2.3 ± 1.0	2.4 ± 0.6	3.3 ± 1.3
	PO_4_^3−^ (µM)	0.4 ± 0.2		0.3 ± 0.0		0.4 ± 0.0		0.4 ± 0.1	0.4 ± 0.1	0.5 ± 0.1
Sediment	TC (%)	8.4 ± 0.9 A		8.3 ± 1.2 A		8.2 ± 1.0 A		4.2 ± 0.3 B	5.7 ± 0.7 B	4.7 ± 0.5 B
	TN (%)	0.1 ± 0.0		0.1 ± 0.0		0.1 ± 0.0		0.1 ± 0.0	0.1 ± 0.0	0.1 ± 0.0
	δ^13^C (‰)	−16.8 ± 0.8 Aa		−18.4 ± 1.1 Aa		−15.6 ± 0.8 Ab		−24.4 ± 0.8 Ba	−20.7 ± 0.9 Bb	−22.5 ± 0.4 Ba
	δ^15^Ν (‰)	1.6 ± 0.8 Aa		0.3 ± 0.1 b		0.6 ± 0.0		−0.9 ± 0.3 B	0.1 ± 0.2	0.0 ± −0.2
	C/N	317.0 ± 6.0		318.7 ± 6.0		406.1 ± 8.0 A		301.5 ± 75.0	199.6 ± 34.5	213.4 ± 38.5 B
Sediment granulometry										
Size (mm)	**>1000**	**1000–250**	**250–125**	**125–63**	**<63**	**>1000**	**1000–250**	**250–125**	**125–63**	**<63**
(%)	16.4	71.5	4.8	4.0	3.4	14.1	74.7	10.7	0.5	0.1

**Table 3 plants-12-00350-t003:** Characterization of biotic descriptors (mean values ± standard deviation for biochemical data). Significant differences between sites (two-way ANOVA and Tukey’s pairwise *post-hoc* test) are indicated in capital letters, in lowercase between meadow zones, and in italics lowercase letter between species.

	SITE #1—Shallow				SITE #2—Deep			
Species	*C. nodosa*		*H. stipulacea*		*C. nodosa*		*H. stipulacea*	
Meadow	Monospecific	Mixed	Monospecific	Mixed	Monospecific	Mixed	Monospecific	Mixed
Shoot density (m^−2^) SE	2059.80 413.0 Aa	486.9 114.0 Ab	3497.6 554.0	4640.0 1028.0	306.4 33.0 B	134.0 47.0 B	3017.0 209.0	2960.0 44.5
Area (cm^−2^) SE	9.2 0.3 a	8.4 0.6 b	1.0 0.1	0.7 0.2 A	10.8 0.9 a	8.4 1.5 b	1.4 0.2	1.3 0.1 B
TLR * (m m^−2^) SE	5.9 2.2 Aa	1.8 0.9 Ab	5.2 2.1	6.3 2.4 A	0.4 0.2 B	0.4 0.1 B	3.7 0.8	4.3 0.2 B
**Leaves**								
Area (cm^−2^)	9.2 0.3 a	8.4 0.6 b	1.0 0.1	0.7 0.2 A	10.8 0.9 a	8.4 1.5 b	1.4 0.2	1.3 0.1 B
Chl *a* (mg g^−1^)	0.5 ± 0.0	0.5 ± 0.0	0.5 ± 0.1	0.3 ± 0.12	1.0 ± 0.3	0.5 ± 0.3	0.5 ± 0.2	0.5 ± 0.3
Chl *b* (mg g^−1^)	0.1 ± 0.0 a	0.3 ± 0.1 b	0.3 ± 0.0 a	0.5 ± 0.2 Ab	0.2 ± 0.3	0.2 ± 0.1	0.1 ± 0.2	0.1 ± 0.1 B
Chl *a*/*b* ratio	0.2 ± 0.1	0.1 ± 0.1 A	0.1 ± 0.0	0.2 ± 0.1	0.2 ± 0.1	0.2 ± 0.0 B	0.3 ± 0.1	0.2 ± 0.1
Car (mg g^−1^)	5.0 ± 0.7	2.0 ± 0.00	1.8 ± 0.2	0.4 ± 0.1	4.4 ± 0.	1.3 ± 1.3	4.6 ± 1.2	4.01 ± 1.9
Chl/Car ratio	7.0 ± 4.1	4.2 ± 0.9	6.7 ± 2.4	6.1 ± 0.7	5.9 ± 1.9	2.7 ± 1.2	2.0 ± 0.6	3.5 ± 2.2
Total phenols (mg g^−1^)	3.1 ± 0.5 a	5.7 ± 0.5 Ab	3.0 ± 1.8 Aa	5.8 ± 0.8 Ab	5.6 ± 1.2a	10.1 ± 3.1 Bb	5.67 ± 1.3 Ba	8.52 ± 1.6 Bb
TC (%)	33.8 ± 0.4 Aa	33.2 ± 0.8 Aa	28.7 ± 0.4 Ab	29.1 ± 0.4 Ab	36 ± 0.5 Ba*a*	30.6 ± 0.5 Bb	26.7 ± 0.5 Ba*b*	27.6 ± 1.2 Bb
TN (%)	1.6 ± 0.1	1.4 ± 0.1	2.1 ± 0.1	1.8 ± 0.1	1.3 ± 0.01	1.3 ± 0.1	1.4 ± 0.1	1.5 ± 0.1
C/N	20.1 ± 1.3 A*a*	22.3 ± 0.6 *a*	13.4 ± 0.7 a*b*	13.4 ± 0.7 b*b*	26.0 ± 21.0 Ba*a*	22.3 ± 1.2 b*a*	18.3 ± 0.6 *b*	18.2 ± 0.8 *b*
δ^13^C (‰)	−7.9 ± 0.1 a	−7.3 ± 0.1 b	−8.0 ± 0.2	−8.0 ± 0.2	−7.8 ± 0.1 a	−7.5 ± 0.0 b	−9.0 ± 0.1	−9.9 ± 0.1
δ^15^Ν (‰)	1.2 ± 0.1 a	1.0 ± 0.0A a	1.9 ± 0.2 Aa*b*	2.1 ± 0.3 Ab*b*	1.2 ± 0.1 aa	0.5 ± 0.0 Bb*a*	2.6 ± 0.1 Ba*b*	2.2 ± 0.0 Bb*b*
**Rhizomes**								
TC (%)	31.2 ± 1.0 A	32.9 ± 0.9 A	27.7 ± 1.6 A	29.5 ± 1.2	35.9 ± 0.7 Ba*a*	29.9 ± 2.7 Bb	25.5 ± 1.9 Ba*b*	29.3 ± 0.8 b
TN (%)	0.4 ± 0.1 A*a*	0.4 ± 0.1 A*a*	0.5 ± 0.1 A*b*	0.6 ± 0.0 A*b*	0.7 ± 0.0 B*a*	0.6 ± 0.1 B*a*	0.4 ± 0.0 B*b*	0.4 ± 0.1 B*b*
C/N	68.2 ± 5.6 Aa	72.7 ± 6.2 Aa	48.4 ± 2.3 Ab	47.1 ± 3.1 Ab	50.3 ± 1.8 Ba	46.4 ± 0.1 Ba	58.3 ± 1.0 Bb	59.2 ± 1.7 Bb
δ^13^C (‰)	−8.0 ± 0.1 Aa*a*	−7.4 ± 0.0 Ab*a*	−8.9 ± 0.1 A*b*	−9 ± 0.1 A*b*	−8.1 ± 0.2 Ba*a*	−7.8 ± 0.1 Bb*a*	−9.7 ± 0.1 Ba*b*	−10.4 ± 0.0 Bb*b*
δ^15^Ν (‰)	−0.5 ± 0.0 Aa*a*	0.6 ± 0.0 Ab	0.5 ± 0.4 A*b*	0.5 ± 0.3 A	−0.4 ± 0.1 B*a*	−0.5 ± 0.3 B*a*	0.8 ± 0.1 Ba*b*	0.9 ± 0.0 Bb*b*
**Roots**								
TC (%)	33.8 ± 2.6 a	25.2 ± 1.3	32.8 ± 0.61 ab	30.8 ± 2.3 b	33.5 ± 0.3 a	30 ± 4.2	25.6 ± 2.6 b	28 ± 1.2
TN (%)	0.6 ± 0.0	0.5 ± 0.0 Aa	0.5 ± 0.0 a	2.9 ± 0.5 Ab*b*	0.6 ± 0.0 a	0.5 ± 0.0 Bb	0.4 ± 0.0 a	0.5 ± 0.0 Bb
C/N	48.4 ± 2	47.0 ± 1.1 ba	57.5 ± 7.9 Aa	10.5 ± 0.9 Ab*b*	48.2 ± 1.2 a	51.4 ± 2.7 ba	55.4 ± 1.9 B	53.9 ± 3.8 Bb
δ^13^C (‰)	−8.2 ± 0.2 a	−8.0 ± 0.0 b*a*	−7.6 ± 0.2 Aa	−8.4 ± 0.0 Ab*b*	−8.1 ± 0.2 a*a*	−7.9 ± 0.1 b*a*	−9.2 ± 0.0 Ba*b*	−9.7 ± 0.0 Bb*b*
δ^15^Ν (‰)	1.1 ± 0.1 A*a*	0.7 ± 0.0 A	0.8 ± 0.1 a*b*	1.1 ± 0.0 Ab	0.8 ± 0.0 Ba	0.9 ± 0.0 Bb*a*	0.8 ± 0.0 a	1.9 ± 0.0 Bb*b*

* TLR = Total length of rhizomes, including branches.

**Table 4 plants-12-00350-t004:** Comparison of the seagrass-associated bacterial communities between seagrass species or meadow zones (two-way ANOSIM).

		Comparison (Two-Way ANOSIM)			
Site	Plant Part	Seagrass Species		Meadow Zone	
		R	*p*	R	*p*
Shallow	Aboveground	0.40	<0.01	0.15	NS
	Belowground	0.35	<0.01	0.07	NS
Deep	Aboveground	0.27	<0.05	0.22	<0.05
	Belowground	0.29	<0.05	0.15	NS

**Table 5 plants-12-00350-t005:** Bacterial taxa that significantly characterize the sediment of each meadow zone, and their putative metabolic traits or functions.

Bacterial Taxon	Putative Metabolic Traits or Functions	References
**Shallow *C. nodosa* sediment**		
Flavobacteriaceae, Pseudoalteromonadaceae	Strictly or facultative anaerobic, involved in sulfur and nitrogen cycling	[70,71]
Lentimicrobiaceae	Involved in sulfur cycling	[72]
**Deep *C. nodosa* sediment**		
Thermoanaerobaculaceae	Nitrogen cycling	[73,74]
Vibrionaceae	Degradative bacteria, associated with seagrass decline	[75,76]
Alteromonadaceae	Degradative bacteria	[77,78]
**Shallow and deep *H. stipulacea* sediment**		
No host-specific associations		
**Shallow and deep mixed sediment**		
Fusobacteriaceae, Izemoplasmataceae	Extracellular DNA degradation	[79,80]

**Table 6 plants-12-00350-t006:** Host-specific bacterial partners significantly associated with aboveground or belowground versions of each seagrass in the two sites, and their putative functions.

Bacterial Taxon	Putative Metabolic Traits and Functions	References
**Shallow *C. nodosa* aboveground—monospecific**		
Micavibrionaceae	Predators of other microbes, including pathogens	[81]
Yersiniaceae	Plant growth promoters	[82,83]
Xenococcaceae, Synechococcales_Incertae_Sedis	Microbes involved in nitrogen cycle	[84]
Parvularculaceae, Oceanospirillales	Halotolerant bacteria associated with other marine organisms	[85,86]
**Shallow *C. nodosa* aboveground—mixed**		
Flavobacteraceae, Moraxellaceae	Complex carbon compounds (including phenols) degraders	[87,88]
Flavobacteraceae	Producer of compound against dinoflagellates	[89]
Moraxellaceae, Pirellulaceae (Planctomycetes)	Biosurfactant producers	[90,91,92]
Phormidesmiaceae (Cyanobacteria)	Diazotrophs	[92]
**Shallow *H. stipulacea* aboveground—monospecific**		
Phormidiaceae, Microcystaceae	Nitrogen fixing Cyanobacteria	[93]
Stappiaceae	CO_2_ fixing and nitrifier bacteria	[94]
**Shallow *H. stipulacea* aboveground—mixed**		
Propionibacteriaceae (Actinobacteria)	Antibacterial compounds producers	[95]
**Deep *C. nodosa* aboveground—monospecific**		
Rhodobacteraceae	Biofilm producers	[96,97]
Nostocaceae (Cyanobacteria)	Nitrogen fixing bacteria	[59,85,98]
Methylophilaceae	Methanol degraders	[99]
Saprospiraceae	Complex carbon compounds degraders	[100]
**Deep *C. nodosa* aboveground—mixed**		
Bacteroidetes	Biofilm producers found in seagrass roots	[101]
Marinibiliaceae	Heterotrophic fermentative metabolism	[102]
Pseudoalteromonadaceae, Myxococcaceae	Anti-bacterial, bacteriolytic and antiviral producers	[103,104]
Colwelliaceae	Nitrogen fixing bacteria	[105]
Thioalkalispiraceae	Sulfur oxidizers	[106]
Hyphomonadaceae	Nitrate reducing	[107]
**Deep *H. stipulacea* aboveground—mixed**		
Microcystaceae, Nisaeaceae, Phormidiaceae (Cyanobacteria)	Nitrogen fixing bacteria	[10,94]
Methyloligellaceae	Methylotrophs Nitrogen fixer	[108]
Desulfocapsacaceae	Sulfur compound reducers	[109]
Pirellulaceae	Leaves colonizers	[90]
**Shallow *C. nodosa* belowground—monospecific**		
Cellvibrionaceae	Nitrogen fixer, growth promoter, lignocellulose degrader	[110,111]
Methylophagaceae	Plant hormones producers, methanol degraders	[112,113,114,115,116]
Bacteroidia	Root associated bacteria	
**Shallow *C. nodosa* belowground—mixed**		
Saccharospirillaceae	Lignocellulose degraders	[117]
Marinibilaceae	Root associated, complex organic materials degraders	[114]
Desulfosarcina	Sulfur compound reducers, root associated bacteria	[22,111]
Sedimenticolaceae	Sulfur-oxidizing bacteria	[115,113]
Bacteroidetes_BD2-2	Root associated bacteria	[116]
**Shallow *H. stipulacea* belowground—monospecific**		
Nodosilinaceae	Nitrogen fixing bacteria	[108]
**Shallow *H. stipulacea* belowground—mixed**		
No association		
**Deep *C. nodosa* belowground—monospecific**		
Saccharospirillaceae	Lignocellulose degraders	[117]
Sphingomonadaceae	Aerobic chemoheterotrophs	[112]
Caulobacteraceae, Alteromonadaceae, Flavobacteraceae	Facultative anaerobic degraders	[78,88,118]
Xanthomonadaceae	Potential plant pathogens	[119]
Rhizobiaceae, Bacteroidetes_BD2-2, Spirochaetaceae,	Root associated bacteria	[120]
Burkholderiaceae		[116]
Flavobacteraceae	Rhizosphere microbes of successfully restored seagrass	[113]

## Data Availability

Genomic data can be found in GenBank PRJNA826444. Data related to the biotic and abiotic setting (collected under an ASSEMBLE Plus Project) have been submitted to the Integrated Marine Information System and can be requested from Chiara Conte at chiara.conte@alumni.uniroma2.eu. Environmental data of the geographical area were taken from: EMODnet Chemistry data—Data and products on marine water quality were provided by the European Marine Observation and Data Network (EMODnet). ICES Oceanographic database was provided by the International Council for the Exploration of the Sea (ICES).

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
