# Peer review of "A Tight Interaction between the Native Seagrass Cymodocea nodosa and the Exotic Halophila stipulacea in the Aegean Sea Highlights Seagrass Holobiont Variations"

_plants, 2023, doi:10.3390/plants12020350_

Round 1

Reviewer 1 Report

This article reported new findings concerning interspecific interactions between two Mediterranean seagrass species from the biometric, biochemical, and microbiome perspectives. Field and laboratory methods used in this study are generally sound, and presentation and interpretation of obtained data are reasonable. I agree that this article should be published in the journal if the authors can successfully address questions and comments I raised below.

(1) The authors defined a ubiquitous component of the epiphytic bacterial community of seagrasses, common to different seagrass species and present in all plant parts and sites, as "bacterial core", which they assumed to be involved in constitutive seagrass processes (Section 3.2). The other component of the bacterial community, i.e., "host-specific component" was assumed to make a conditional and temporary association with the host. I think that they should explain why they interpreted like that, because it is intuitively not trivial. Rather, it seems to me more natural to assume that the host-specific component consists of bacteria tightly integrated into the holobiont as a result of co-evolution, while the ubiquitous bacteria may be seen as common environmental bacteria living in water column or sediment and interacting with the host only opportunistically.

(2) Bacterial community composition in this study was evaluated and compared taxonomically at the family level (Section 4.4). I agree that this is a reasonable way for agglomeration. However, by doing so, important differences of species composition (i.e. at below-family levels) may be obscured and a fake similarity between different communities may be concluded by, e.g., PCoA analysis like Figure 1. Such results should be treated with reservations. In the classical ecological context, the concepts of similarity and diversity of communities are meaningful only when they are evaluated at the species level. So, the biological/ecological significance of the concepts such as alpha-diversity at the family level may be difficult to define. Can the authors explain it, or cite appropriate references?

(3) The authors should explain in detail the method for isolating epiphytic bacteria from above- and below-ground components of seagrasses for the amplicon sequencing (Section 4.3). They just described that "The tissue were carefully rinsed with 2 mL of a washing solution..., the solution centrifuged ... and the pellet stored". However, epiphytic bacterial community potentially consists of various components with different strength of attachment, and it should be considerably difficult to extract all of such subcommunities with an equal recovery. Even if the same washing solution is used for the same sample, the composition of the final isolates may differ significantly case-by-case. How could they ensure that the quality of the extracts was essentially same for all the samples? Have they checked the reproducibility?

(4) The authors measured stable C and N isotope ratios of seagrass tissues but did not mention them at all in Discussion. I think the results (Table 3) are quite interesting and should be elaborated. For example, the d13C of Cymodocea was always higher than that of Halophila, suggesting that the latter depended more extensively on CO2 derived from sediment respiration as opposed to water-column DIC. The difference became even greater in the mixed beds, which indicates that Halophila was a superior competitor over Cymodocea in acquiring CO2 rapidly. The d15N of seagrass leaves, on the other hand, was always higher than rhizomes and roots, suggesting that DIN in the water column had a higher d15N than DIN in pore water. The d15N of Halophila leaves was even higher than Cymodocea leaves in the mixed beds, which suggests that the former could absorb water column DIN more efficiently than the latter.

Author Response

The Authors wish to thank the Referee #1 for the kind and constructive evaluation of their work. All the comments were cautiously considered and we tried to answer at our best to the points indicated by the Referee, as reported below point by point.

Referee comment #1:

The authors defined a ubiquitous component of the epiphytic bacterial community of seagrasses, common to different seagrass species and present in all plant parts and sites, as "bacterial core", which they assumed to be involved in constitutive seagrass processes (Section 3.2). The other component of the bacterial community, i.e., "host-specific component" was assumed to make a conditional and temporary association with the host. I think that they should explain why they interpreted like that, because it is intuitively not trivial. Rather, it seems to me more natural to assume that the host-specific component consists of bacteria tightly integrated into the holobiont as a result of co-evolution, while the ubiquitous bacteria may be seen as common environmental bacteria living in water column or sediment and interacting with the host only opportunistically.

Authors reply:

The bacteria found on both species are likely to colonize the seagrass surface due to their wide distribution and the surface adapted lifestyle, as mentioned in the text. Nevertheless, the entity of colonization, higher on the seagrass than in environmental pool suggests that they may find a suitable micro-niche on seagrass, as also suggested by the literature data. The repeated findings in the literature also support this hypothesis. This does not exclude that some families may be opportunistic bacteria, too. This point has been added at lines 406-412.

As regard the host-specific component of the bacterial community, it has been related to the host physiology, due to their variable occurrence in different conditions. This point has been clarified at lines 413-417.

As a last point, the functional relationship and, particularly, the coevolution among plants and their epiphytic bacteria is a very fascinating issue. Unfortunately, at this phase of the studies we do not have enough data to demonstrate such a complex phenomenon. At the moment there is a methodological gap to fill: our data (and conclusions) come from the evaluation of community structures and not actual functions; when it will be possible to study the functionalities of the two partners simultaneously, then it will be possible to go into more detail about this phenomenon and to give a deeper insights into the plants-bacteria relationship. Being too far from our actual capacity, the point has not been added in the discussion of the data.

Referee comment #2:

Bacterial community composition in this study was evaluated and compared taxonomically at the family level (Section 4.4). I agree that this is a reasonable way for agglomeration. However, by doing so, important differences of species composition (i.e. at below-family levels) may be obscured and a fake similarity between different communities may be concluded by, e.g., PCoA analysis like Figure 1. Such results should be treated with reservations. In the classical ecological context, the concepts of similarity and diversity of communities are meaningful only when they are evaluated at the species level. So, the biological/ecological significance of the concepts such as alpha-diversity at the family level may be difficult to define. Can the authors explain it, or cite appropriate references?

Authors reply:

We completely agree with the referee comment, for this reason the statistical analyses were performed using the normalized Amplicon Sequence Variant (AVS) dataset, and AVS give indication of even slight differences among the different DNA collected in the field, at a finer scale than family. Conversely, the dataset agglomerated at the family level was used to visualize and describe the bacterial communities in a quite simple way. This choice and procedure has been clarified in the text at lines 650-652.

Referee comment #3:

The authors should explain in detail the method for isolating epiphytic bacteria from above- and below-ground components of seagrasses for the amplicon sequencing (Section 4.3). They just described that "The tissue were carefully rinsed with 2 mL of a washing solution..., the solution centrifuged ... and the pellet stored". However, epiphytic bacterial community potentially consists of various components with different strength of attachment, and it should be considerably difficult to extract all of such subcommunities with an equal recovery. Even if the same washing solution is used for the same sample, the composition of the final isolates may differ significantly case-by-case. How could they ensure that the quality of the extracts was essentially same for all the samples? Have they checked the reproducibility?

Authors reply:

Our group of research perform this kind of sampling since 2014. Before the first publication (Mejia et al., 2016) in which the method has been defined, several trial tests have been done to define the most efficient protocol. Of the tested sampling methods the one described in Mejia et al., 2016 was the most efficient and has been then applied in all the following sampling campaigns. Furthermore, in all the sampling campaigns, we are used to collect 5 samples for each experimental batch, four of them are extracted and the amount of DNA measured; then the three best are sent to sequencing. In any case, we send to sequencing set of samples, all with a comparable amount of DNA, checked by PCR. The efficacy of the method came clear from both the rarefaction curves of all the studies, and by the comparison with the results of other studies. The minimum amount of DNA in each replicate sample sent to sequencing (≥ 5 ng µl-1) has been added at line 606.

Referee comment #4:

The authors measured stable C and N isotope ratios of seagrass tissues but did not mention them at all in Discussion. I think the results (Table 3) are quite interesting and should be elaborated. For example, the d13C of Cymodocea was always higher than that of Halophila, suggesting that the latter depended more extensively on CO2 derived from sediment respiration as opposed to water-column DIC. The difference became even greater in the mixed beds, which indicates that Halophila was a superior competitor over Cymodocea in acquiring CO2 rapidly.

The d15N of seagrass leaves, on the other hand, was always higher than rhizomes and roots, suggesting that DIN in the water column had a higher d15N than DIN in pore water. The d15N of Halophila leaves was even higher than Cymodocea leaves in the mixed beds, which suggests that the former could absorb water column DIN more efficiently than the latter.

Authors reply:

We thank the Referee for the suggestion to include more details about the stable isotopes; we agreed and, to this end, we included the section reported below at lines 378-393 and the new references into the Reference section.

The values and the differences of carbon content between seagrasses found in this study, lined within the natural range and the difference depends on interspecific variation [135, 136]; this indicated that plants maintain their physiological features, in spite of the different sediment carbon content. Conversely, the isotopic content indicated a different growth dynamics of the two species and gave an insight into their dynamics under competition. In fact, C. nodosa in both sites and meadow zones always contained higher concentration of δ13C‰ than H. stipulacea. These differences may depend on a higher efficiency of H. stipulacea in absorbing light carbon isotope due a fast growth rate (0.2 cm d−1 [36] for Halophila vs 0.004 day-1 for Cymodocea [136]), this may represent a potential advantage of the exotic on the native seagrass, especially in nutrient limiting condition [64]. A different dynamics has been found for nitrogen: in the sediment of the two sites total nitrogen did not showed differences, but it changed in plants tissues. This is probably due to the associated N-fixing bacteria, able to supply plants with the vital nutrient [13]. However, in the mixed zone the δ15N content was higher in H. stipulacea than in C. nodosa leaves, this suggests that the exotic species must be able to absorb nitrogen from other matrices, i.e. from water column, in a more efficient way than the native one.

Duarte, C. M., Sand-Jensen, K. Seagrass colonization: patch formation and patch growth in Cymodocea nodosa. Mar. Ecol. Progr. Ser. 1990, 65, 193-200.

Azcárate-García, T., Beca-Carretero, P., Villamayor, B., Stengel, D. B., Winters, G. (2020). Responses of the seagrass Halophila stipulacea to depth and spatial gradients in its native region (Red Sea): morphology, in situ growth and biomass production. Aquat. Bot. 165, 103252.

Campbell, J.E., Fourqurean, J.W. Interspecific variation in the elemental and stable isotope content of seagrasses in South Florida. Mar. Ecol. Progr. Ser. 2009, 387, 109-123. https://doi.org/10.3354/meps08093

Reviewer 2 Report

Review of the manuscript Plants-2118077 A tight interaction between the native seagrass Cymodocea nodosa and the exotic Halophila stipulacea in the Aegean Sea highlights seagrass holobiont variations written by Chiara Conte, Eugenia T. Apostolaki, Salvatrice Vizzini and Luciana Migliore

This paper presents the evaluation of competition between exotic seagrass Halophila spipulacea and native Cymodocea nodosa in the Aegean Sea, by analysing epiphytic and rhizophytic bacterial communities of two species in three zones (mixed, and monospecific for each of two species) at two depths. The evaluation of the differences between holobionts were based on seagrass descriptors (morphometric, biochemical, elemental and isotopic composition) and 16S rRNA gene analysis. The native C. nodosa was affected by the presence of exotic H. stipulacea, while H. stipulacea remained almost unchanged. This study provided evidence of the competitive advantage of H. stipulacea on C. nodosa in the Aegean Sea and suggests the possible use of associated bacterial communities as a further descriptor of native seagrasses sustainability.

Review of the manuscript Plants-2118077 A tight interaction between the native seagrass Cymodocea nodosa and the exotic Halophila stipulacea in the Aegean Sea highlights seagrass holobiont variations written by Chiara Conte, Eugenia T. Apostolaki, Salvatrice Vizzini and Luciana Migliore

This paper presents the evaluation of competition between exotic seagrass Halophila spipulacea and native Cymodocea nodosa in the Aegean Sea, by analysing epiphytic and rhizophytic bacterial communities of two species in three zones (mixed, and monospecific for each of two species) at two depths. The evaluation of the differences between holobionts were based on seagrass descriptors (morphometric, biochemical, elemental and isotopic composition) and 16S rRNA gene analysis. The native C. nodosa was affected by the presence of exotic H. stipulacea, while H. stipulacea remained almost unchanged. This study provided evidence of the competitive advantage of H. stipulacea on C. nodosa in the Aegean Sea and suggests the possible use of associated bacterial communities as a further descriptor of native seagrasses sustainability.

The topic of the paper is very interesting. The experiment was set up very meaningfully to ensure good comparison and statistical support of the results. Based on the valuable and large data-set, this paper provides new important information about the holobionts and interactions between exotic and native seagrasses in the areas where two species coexisting.

Materials and Methods are sufficiently detailed and straightforward.

The results were conclusive, supported by the statistical analysis, and well presented.

The discussion is summarizing giving a good synthesis of results.

Throughout the references: the journal volume should be put in italic (but check with Guide for authors).

I made some corrections and suggestions in the manuscript which is attached, and denote the places were clarification is needed. Please note that I am not a native speaker.

Author Response

The Authors are really grateful to Referee #2 for the interesting and insightful comments and for the help to ameliorate the text (none of us is mother tongue, too!). We incorporated all the Referee suggestions in the text and completely revised the reference section.

Referee comment #1:

Throughout the references: the journal volume should be put in italic (but check with Guide for authors).

Authors reply:

Done, thank you.

Referee comment #2:

I made some corrections and suggestions in the manuscript which is attached, and denote the places were clarification is needed. Please note that I am not a native speaker.

Authors reply:

Thank you, all the suggestions/corrections were incorporated into the text.

Reviewer 3 Report

The problem under review is interesting when seen from the standpoint of the general ecology of the relationship between different types of algae. The authors interestingly and correctly formulate the task and its implementation. The article contains a lot of factual material. Articles of this type, I believe,should be welcomed and widely published. This article will be of interest to a wide range of researchers. It may be published in the form it is submitted to the editors, but I invite the authors to lend an ear to my comments.

Abstract:

In the abstract, the problem is well formulated, the subject and object of research being clearly defined, but the results of the work are not fully described. It is necessary to expand them and name/list the highlights of the work.

Introduction:

The introduction is well organized and I have no comments to make on this section. Yet, in reference with the following text (lines: 38 - 40): "All organisms, including seagrasses, are colonized by different microbial 38 communities, such as bacteria, fungi, and viruses, constituting a functional unit called 39 holobiont [1-3]," I propose adding the following source: “Khailov K.M. Ecological metabolism in the sea. Naukova Dumka, Kyiv. 1971, 228 p. (in Japanese)." This book focuses on the external metabolic relationships of different groups of organisms, including bacteria and unicellular and multicellular organisms. The above-mentioned book was the first to dwell in detail upon the problem you discuss in your article, and definethe main positions and concepts. This said work was published in 1971 in three languages: English, Russian and Japanese. I have an opportunity to hold a Japanese version in my hands, and therefore I recommend this source for citing.

Results:

The Results section is well structured and commendable for its contents. Yet, I have a small remark to make here. Section “2.1. Biotic setting” provides, generally, a qualitative parameterdescription. In my view, if the parameter assessment is accompanied by quantitative data, indicating how much the characteristic under comparison gains or loses in value, the quality of the information presented will increase. In Table 3, this information is present, but not explicitly.

Discussion.

Lines: 354 - 357. You use the term "competition". It remains unclear to me the use of this term in relation to your tasks. What kind of resource is being competed for?

Lines: 361 – 363. I failed to find the work [126] and, accordingly, to get acquainted with the statement of its authors. Could you please specifythe competition mechanism you observe?Is it a competition for light or for growth conditions in the ground?

Lines: 365 – 370. No doubt, the author of this article as well as the authors of [68, 127], who are closely acquainted with this problem, understand clearly the mechanism of competition between C. nodosa and H. Stipulacea, but it is not clear to me as a reader why C. nodosa loses to H. stipulacea. It would seem that C. nodosa forms a high canopy and can successfully compete for light.

Materials and Methods

Lines: 365 - 370. In my opinion, the article would gain a lot if you specify the width of the mixed zone.

Author Response

The Authors appreciated the Referee #3 constructive suggestions and cautiously considered and addressed all the comments. The details are reported in the following point by point reply to the reviewer comments.

Referee comment #1:

In the abstract, the problem is well formulated, the subject and object of research being clearly defined, but the results of the work are not fully described. It is necessary to expand them and name/list the highlights of the work.

Authors reply:

Thank you for the comments, we expanded the Results section according to the request, as also reported in the reply to points #3 and #4. 

Referee comment #2:

I propose adding the following source:  “Khailov K.M. Ecological metabolism in the sea. Naukova Dumka, Kyiv. 1971, 228 p. (in Japanese)

Authors reply:

We are very very sorry, but the suggested reference has not been added, because we have desperately searched for the book in these days, but it seems to be impossible to find. Without having read any part of the text, we have no way of adequately citing the book in the text.

Referee comment #3:

Lines: 354 - 357. You use the term "competition". It remains unclear to me the use of this term in relation to your tasks. What kind of resource is being competed for?

Authors reply:

Due to the different growth rate and structure, the main resource the two species compete for is space, that obviously imply the availability of many other resources, including nutrients and light; Halophila, due to its faster growth rate can easily ‘occupy’ space, hindering the growth of the native species. One aspect of this Halophila fast colonization has been clarified at line 384-386: isotopic data suggest that Halophila is a better competitor as regards the taking up of nutrients, as it absorbs more effectively the light carbon isotope – more easily absorbed – leaving to Cymodocea the heavy one. Furthermore, as reported in the next point, the H. stipulacea high plasticity and growth rate make the plants able to face the shading of high canopy plants, as C. nodosa: the leaves become larger under lower light conditions. Hence, the morphological plasticity together with the fast growth rate of Halophila plants makes this species a good competitor even against taller plants, as Cymodocea. These concepts have been added; the amendments are reported at lines 395-402.

Referee comment #4:

Lines: 365 – 370. No doubt, the author of this article as well as the authors of [68, 127], who are closely acquainted with this problem, understand clearly the mechanism of competition between C. nodosa and H. stipulacea, but it is not clear to me as a reader why C. nodosa loses to H. stipulacea. It would seem that C. nodosa forms a high canopy and can successfully compete for light.

Authors reply:

The referee is right while saying that the high canopy of C. nodosa may shade H. stipulacea plants while in syntopy. Nevertheless, as already reported in point #3, the H. stipulacea high morphological plasticity and growth rate makes the plants able to respond to this damage: the leaves become larger under low light conditions and its fast growth rate make this species a good competitor even against taller plants, as indicated by the different heavy C isotope content found in the two seagrasses in the mixed zones. This part has been completely rewritten and the concept clarified in lines 395-402.

Referee comment #5:

In my opinion, the article would gain a lot if you specify the width of the mixed zone.

Authors reply:

In the text the dimensions of the mixed zone have been included, the added text (lines 493-496) is the following:

‘In both sites the mixed zones were smaller than the monospecific ones. In the shallow site the mixed meadow covered approximately an area of 30-40 m2, while in the deep site the extension was about 20-30 m2.'